# Short-term coastal forest responses to a hurricane-scale freshwater and saltwater flooding experiment

Allison N. Myers-Pigg[1,2☯*], Anya Hopple[3☯], Stephanie C. Pennington[4], Peter Regier[1], Ben Bond-Lamberty[4], Mia J. DiCianna[3], Kennedy O. Doro[2], Nate McDowell[5], Julia McElhinny[1], Alice Stearns[3], TEMPEST 1.0 Event Consortium[¶], Nicholas D. Ward[1,6], Vanessa L. Bailey[2,5], J. Patrick Megonigal[3,5*]

1 Pacific Northwest National Laboratory, Sequim, Washington State, United States of America,
2 University of Toledo, Toledo, Ohio, United States of America, 3 Smithsonian Environmental Research Center, Edgewater, Maryland, United States of America, 4 Joint Global Change Research Institute, Pacific Northwest National Laboratory, College Park, Maryland, United States of America, 5 Pacific Northwest National Laboratory, Richland, Washington State, United States of America, 6 University of Washington, Seattle, Washington State, United States of America

¶ Complete membership of TEMPEST 1.0 Event Consortium is provided in the Acknowledgements
☯ These authors contributed equally to this work
* allison.myers-pigg@pnnl.gov (ANMP); megonigalp@si.edu (JPM)

## Abstract

Coastal upland forests are exposed to intensifying precipitation regimes and sea level rise, increasing tree mortality and transforming these coastal forests into wetland ecosystems. While the ultimate outcome of long-term exposure to these perturbations is known to be an ecosystem state change from upland forest to wetland, the resistance of forests to the first novel exposure to flooding and salinity is relatively unknown. The Terrestrial Ecosystem Manipulation to Probe the Effects of Storm Treatments (TEMPEST) experiment uses ecosystem-scale (2000 m$^2$) experimental flooding plots to decouple two distinct disturbances associated with hydrological extremes: (1) freshwater saturation of soils and flooding (e.g., from heavy precipitation) and (2) salinization from storm surge by saturating and flooding soils with brackish water. Here we describe the immediate effects of the experimental flooding treatments on hydrologic, biogeochemical and vegetation ecosystem components following the first novel experimental ecosystem-scale flooding event in TEMPEST. Following a 9-hour experimental treatment, the system's hydrology was temporarily and significantly impacted, but there were subtle effects on biogeochemical and vegetation components of the ecosystem. This suggests that this temperate deciduous forest was resistant to a single novel flooding event, even if the water is saline. Most biogeochemical parameters monitored in the soil, porewater, and groundwater responded similarly between freshwater and saltwater treatments relative to the control plot. However, we show that even a single episodic event can cause large transient shifts in belowground conditions that drive physiological changes in coastal forest functions, such as soil

**Data availability statement:** All data and code are freely available on ESS-DIVE and accessible at [28].

**Funding:** This research was supported by COMPASS-FME, a multi-institutional project supported by the U.S. Department of Energy, Office of Science, Biological and Environmental Research as part of the Environmental System Science Program, and by the Smithsonian Environmental Research Center. The funders had no role in study design, data collection and analysis, decision to publish, or preparation of the manuscript.

**Competing interests:** The authors have declared that no competing interests exist.

moisture and oxygen levels. Such responses may impact how the system responds to future perturbations.

## Introduction

The upland boundaries of coastal ecosystems are becoming increasingly exposed to hydrological extremes such as storm surge, coastal flooding, extreme rain events, and hurricanes, causing ecophysiological stress that over time can dramatically transform upland forests through increased tree mortality to 'ghost forests' and then wetlands [1]. Shifts in inundation and saltwater exposure dynamics drive changes beyond just vegetation community composition [1], including key indicators of ecosystem health, such as net ecosystem production [2], carbon sequestration potential [3,4], and soil microbial activity [5]. While thresholds of flooding and salinity exposure associated with coastal forest mortality have been observed [6–8], the underlying mechanisms remain unclear. Hypothesized mechanisms involve a complex mixture of above and belowground processes [9], which may even be triggered by a single, novel salinity exposure [10]. Changes in soil water content, conductivity, and dissolved oxygen content can alter belowground biogeochemical processes and drive vegetative stress [9,11,12]. The effects of flooding and salinity exposure on biogeochemical and ecological processes may be immediate, driven by shifts in the availability of oxygen and key substrates [13,14], or lagged, as plant or soil microbial communities become more stressed over time [9].

Acute pressures (e.g., pulse disturbances) may produce biogeochemically important, yet transient, effects [15] that nonetheless shape the trajectories of long-term aboveground [9,16] and belowground [11,17] coastal forest responses to subsequent events. In particular, sites experiencing infrequent or new exposure to salinity may be more sensitive than sites with more frequent exposures [11]. Previous studies in systems with episodic historical exposure to salinity are consistent with this idea, finding that chronic episodic salinity exposure events caused the most dramatic changes in ecosystem functions such as soil greenhouse gas production, while acute treatments did not result in sustained shifts [15]. Pulse flooding events can rapidly increase soil saturation and decrease oxygen availability on hourly timescales, driving shifts in the microbial metabolism underpinning soil greenhouse gas production [18–20]. Soil transplant and mesocosm studies find that antecedent conditions effectively regulate belowground biogeochemical responses to flooding and salinity [11,17,18,21,22]. In particular, responses of soil carbon solubility and microbial respiration to salinity are linked to antecedent conditions and soil properties [22]. Likewise, exposure to hypoxia and increased salinity drive root damages and mortality [9]. A single, novel disturbance could therefore pre-dispose an ecosystem to be less resistant to future disturbance events by altering the drivers of a variety of above and belowground biogeochemical processes, thus changing antecedent system conditions for future events [17].

Here we consider the effects of novel flooding on a coastal upland forest by experimentally flooding large forest plots (2000 m$^2$) with freshwater and saline water to achieve transient soil saturation [23]. The objectives of the present study are to

determine if the first novel exposure impacted key hydrologic, biogeochemical, and vegetation response variables that are mechanistically linked to above and belowground ecosystem functions (Table 1). We focus on a suite of variables that capture the mechanistic cascade by which flooding and salinity alter tree and soil functions [9]. In this framework, soil saturation alters oxygen availability, potentially reshaping belowground microbial processes, root processes, and tree physiological functions (Table 1). We chose the variables in this study, such as soil oxygen levels, on the assumption that they represent early indicators of plant or microbial stress [24]. We include both driver (soil and groundwater water content, conductivity, dissolved oxygen) and response variables (microbial and root greenhouse gas fluxes, soil porewater carbon concentration and composition, and tree sap flow velocity); the full list of variables and how they are linked is detailed in Table 1. We hypothesize that hydrologic state variables (e.g., soil moisture, groundwater levels) will respond immediately but transiently to flooding, followed by more muted biogeochemical responses (e.g., soil greenhouse gases, soil porewaters), and minimal vegetation responses (e.g., tree sap flow velocity) within 48 hours. We further hypothesize that hydrologic responses will be similar between freshwater and saline flooding, but that salinity exposure will initiate stronger biogeochemical and vegetative responses than freshwater flooding over the same period. By isolating ecosystem responses during and immediately following the first, novel flooding event, we present short-term shifts that may occur – establishing antecedent conditions that could shape future responses (Fig 1).

## Methods

### Site description

The Terrestrial Ecosystem Manipulation to Probe the Effects of Storm Treatments (TEMPEST) experiment is set in a temperate, deciduous forest on the western shore of Chesapeake Bay in Maryland, USA (38.876°N, 76.553°W) at the

**Table 1. List of response variables considered in this study and their importance to coastal upland forest ecosystem functions.**

| Variable | System Component | Importance |
|---|---|---|
| Soil water content | Hydrological | Water availability is a physiological need of vegetation and microbes but can also cause stress. |
| Soil conductivity | Hydrological | Used as a proxy for soil salinity, which can drive mortality for above and belowground species not tolerant to it. |
| Groundwater depth | Hydrological | Metric of vertical connectivity between surface soils and the groundwater system, an indicator of hydrologic context of the overall system. |
| Groundwater conductivity | Hydrological | Metric of vertical connectivity between surface soils and the groundwater system in response to saline flooding; values should be low in an unperturbed upland forest system. |
| Soil porewater dissolved organic carbon (DOC) | Biogeochemical | Dissolved organic carbon is both a function and driver of many biogeochemical processes, including microbial processing of organic matter in soils. |
| Microbial methane ($CH_4$) flux | Biogeochemical | Metric of microbial greenhouse gas production from soils, in this case without the influence of plant roots. |
| Microbial carbon dioxide ($CO_2$) flux | Biogeochemical | Metric of microbial greenhouse gas production from soils without the influence of plant roots. |
| Soil oxygen ($O_2$) | Biogeochemical | Hypoxic or anoxic conditions can stress vegetation and alter microbial decomposition pathways. |
| Soil porewater spectral slope ratio | Biogeochemical | Indicator of the quality of dissolved organic matter in soil porewaters, and susceptibility to microbial decomposition that can influence greenhouse gas production. |
| Groundwater dissolved oxygen ($O_2$) | Biogeochemical | Hypoxic or anoxic conditions can stress vegetation and alter microbial decomposition pathways. |
| Tree sap flow velocity | Vegetation | Provides rapid insights into how the physiological response of trees to perturbation and overall tree health. |
| Root-influenced $CH_4$ flux | Vegetation | Metric of greenhouse gas production in soils, including the influence of plant roots. |
| Root $CO_2$ flux | Vegetation | Metric of greenhouse gas production by plant roots. |

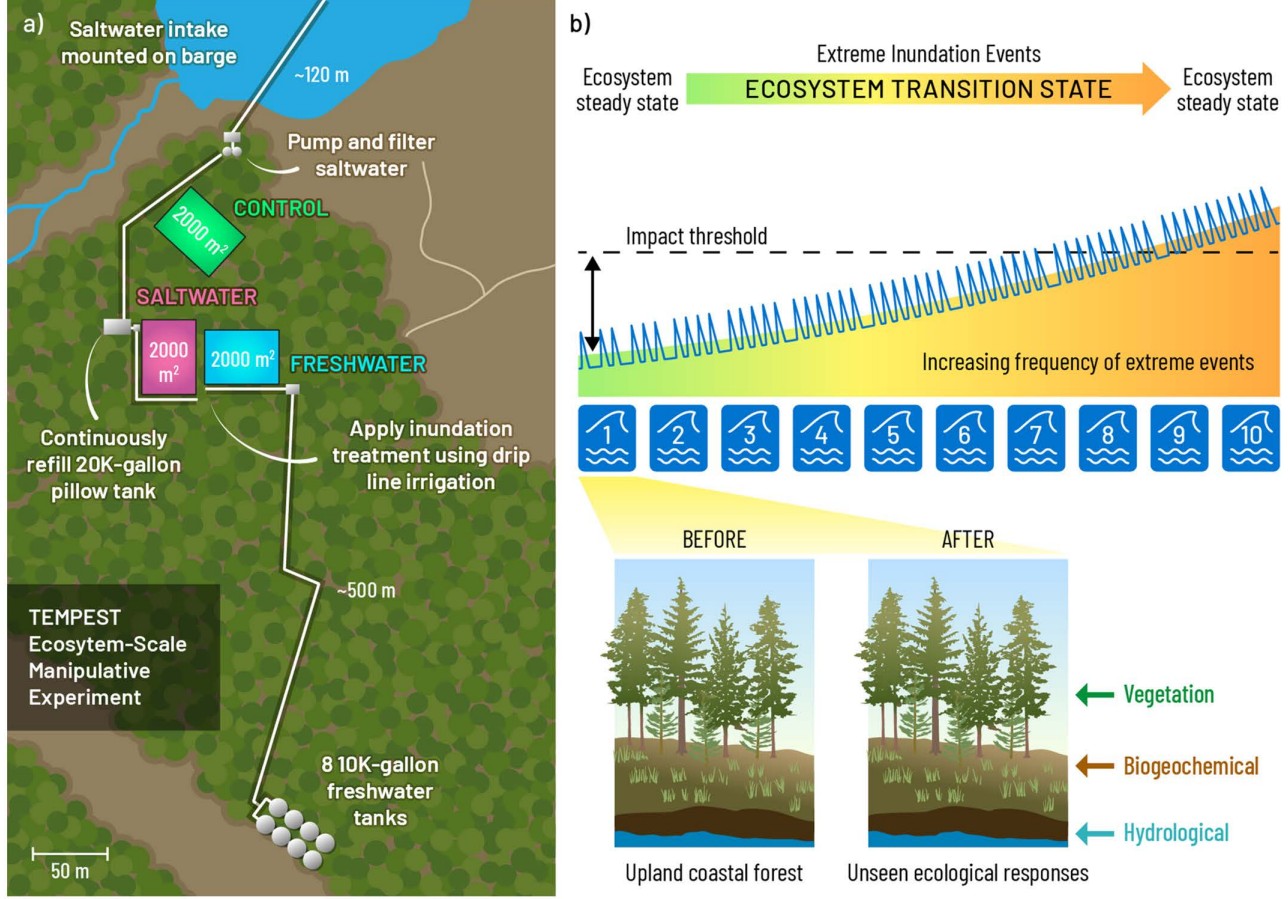

**Fig 1. a) Site and experimental design, inspired by [23]. b) The blue lines and waves represent planned flooding event(s), with the first event described herein denoted as "1".** Each planned event will increase the frequency of flooding, i.e., the second event will flood the plots for 10 hours on each of 2 consecutive days, the third floods the plots for 10 hours on each of 3 consecutive days, etc. Panel b is inspired by [25]. Zooming into the first event described here, we focus on vegetation, biogeochemical and hydrological variables, changes to which are not directly observable aboveground. Note that we employ an adaptive experimental design, where we make necessary changes to infrastructure and timing of events based on what we've learned in previous years. Therefore, the blue waves and lines do not depict the exact details of future events.

Smithsonian Environmental Research Center (SERC: https://serc.si.edu/). The ~60-year-old upland coastal forest is dominated by *Liriodendron tulipifera*, *Fagus grandifolia*, *Acer rubrum*, and *Quercus* spp., while the understory is composed of deciduous shrubs such as *Rubus phoenicolasius*, *Lindera benzoin*, and *Berberis thunbergii* and herbaceous perennials such as *Mitchella repens*, *Polygonum virginianum*, *Rhus radicans*, and *Symphyotrichum lateriflorus*. The water table is on average ~2 m belowground and is drained by a second-order stream that flows into a brackish tidal marsh with a mean annual salinity of 10 psu and tidal range of 44 cm. The soils are well-drained fine sandy loams or sandy loams classified as Typic Hapludults [26]. More details on the soils and vegetation characteristics can be found in [23].

TEMPEST simulates extreme, ecosystem-scale freshwater and saltwater disturbance events using a novel, large-unit (2000 m²), un-replicated experimental design, with three 50 m × 40 m plots serving as control, freshwater, and saltwater treatments (Fig 1). These plots had no known prior exposure to saline conditions. A high-resolution spatiotemporal approach achieved with a grid-system strategy is used to monitor the impacts of experimental treatments on subsurface hydrology, biogeochemistry, and vegetation across these large plots in response to the simulated flooding events. Each

flooding event is targeted to last 10 hours, and each year the number of events is increased (Fig 1). Plots are designed to spatially coordinate measurements spanning the soil-plant-atmosphere continuum. Temporal patterns are captured by continuous sensor networks complemented by discrete measurements at regular bi-weekly or monthly intervals, with higher frequency immediately prior to, during, and following treatment events [27]. Site access was provided by the Smithsonian Institution and a permit to extract the seawater from the Rhode River Estuary was provided by the Maryland Department of the Environment.

### Flood manipulation

Detailed methodology on the flood manipulations can be found in [23]. Briefly, intake systems draw freshwater from a municipal source and brackish water from the adjacent Rhode River estuary, which is then distributed through a network of irrigation tubing equipped with pressure-compensating emitters. The water delivery rate is slightly greater than the drainage capacity of the soil to maximize the time that the soil remains saturated while minimizing water loss by surface runoff. For each flooding event, we aim to deliver 300 $m^3$ of water, ~15 cm, to each 2000 $m^2$ treatment plot at an average rate of 640 L per minute (LPM) over a 10-h period. For simplicity, we refer to the plot where the brackish water treatment was applied as the "saltwater" plot.

### Hydrological, biogeochemical and vegetation data collection

**Soil and groundwater in-situ sensors.** Soil temperature, moisture content, and electrical conductivity (EC) are key drivers of vegetative and microbial stress (Table 1) [9]. Therefore, we measured these variables using TEROS 12 soil sensors (Meter Group) deployed at 5, 15, and 30 cm depths in five grid cells in each plot and deployed at 15 cm in an additional 31 grid cells in each plot (S1 Fig). The sensors were installed in 2020 [23]. Soil oxygen ($O_2$) was measured using Firesting optical oxygen sensors (Pyroscience, Germany) at 30 cm depth in a single grid cell in each plot; sensors were installed prior to the first flood event and removed the day after the event. Groundwater depth, salinity, and dissolved $O_2$ were measured by Aqua TROLL 600 multiparameter sondes (In-Situ) installed in 2019 in ~4 m-deep groundwater wells in a single central grid cell in each plot [23]. A generalized map of sensor installation locations across the experimental plots can be found in the Supporting Information (S1 Fig).

All sensor data were first visually inspected for outliers or sensor malfunctions. TEROS sensor datasets (soil temperature, moisture, and EC) were linearly gap-filled for any missing data (<2% missing for every sensor, with a maximum gap of 2 hours), then binned by timestamp across all sensors for each plot. For Firesting datasets, we replaced all values < 0 with 0, which we suggest are due to small differences (minimum measured value: −0.254 mg/L) between actual 0 and the calibrated value for 0. Groundwater depths below the soil surface were calculated from pressure and well dimensions after correcting for atmospheric pressure and water density. Each of the resulting datasets contained 15-minute time-step resolution data at the individual plot scale.

**Soil porewater.** Soil porewater carbon content and chemistry both drive and respond to alterations in microbial functions in response to salinity and hypoxia (Table 1) [15]. Grab samples of soil porewater were collected for chemical characterization before, during, and after the event from 10 permanently installed lysimeters at 15 cm depth, distributed across each of the plots (S1 Fig). Samples for measuring optical properties of soil porewater were pooled within a given plot due to volume limitations. Samples were field-filtered immediately after collection using a 0.45 µm syringe filter (MilliporeSigma™ Millex™ Nonsterile 33 mm Syringe Filters) and stored at 4 ˚C until analysis. Dissolved organic carbon (DOC) was measured on filtered samples within one week of collection on a Total Organic Carbon Analyzer (Shimadzu TOC-L). DOC was measured as non-purgeable organic carbon (NPOC) via catalytic combustion after in-line acidification with 1:12 hydrochloric acid. Check standards were run every 10 samples. Data underwent additional quality control, including visual inspection of calibration curves, check standards, and sample peak shapes. Peaks were disregarded if the coefficient of variation between replicate injections was > 2.0%, and values were flagged when they were outside of the

calibration curve and instrument detection limit ranges. Further, values were removed from subsequent analysis if dilution factors were > 30, blank values were ≥ 25% of sample values from the instrument, or when replicate samples were > 25% apart. UV absorbance scans and excitation-emission matrices (EEMs) were collected simultaneously on filtered samples using an Aqualog (Horiba Scientific), with absorbance measured from 230 to 800 nm in 3 nm intervals. EEMs were collected within the same wavelength constraints and were further processed with the drEEM toolbox v. 6.0 for Matlab. Absorbance data were blank-corrected prior to exporting data in the Aqualog software. Processing of the EEMs in the drEEM toolbox included blank correction, inner filter correction, and normalization to Raman Scatter units based on daily water Raman scans collected at an excitation of 350 nm.

**Greenhouse gases.** Soil $CH_4$ and $CO_2$ flux are key response variables that can indicate microbial activity in response to salinity exposure and altered redox conditions (Table 1) [17]. Soil $CH_4$ and $CO_2$ flux measurements were taken at permanently installed collars distributed throughout the plots using an infrared gas analyzer (IRGA; LI-7810, LI-COR Inc., Lincoln, NE) attached to a 20 cm-diameter soil flux chamber accessory (LI-8201 Smart Chamber). Samples were collected the day before, during, and for several days after the event [28]. The IRGA measured concentrations every second over a 1-min period and calculated flux based on a nonlinear regression of gas concentration in the closed chamber system over time per unit area. Two successive measurements were taken at each collar and averaged using the LI-COR SoilFluxPro software (v4).

We installed a root-exclusion experiment within each of the three experimental plots to further understand the role of roots on soil $CH_4$ and $CO_2$ emissions [12]. Root emissions are sensitive to salinity and hypoxia-induced vegetative stress (Table 1) [12]. Briefly, we established eight small (0.5 m²) subplots in each of the large experimental plots to impose two treatments, replicated four times each. We trenched four of the subplots to a depth of 60 cm and lined the resulting monoliths with 45 μm mesh to exclude roots alone (i.e., root_free plots) but not mycorrhizae. The other four subplots were undisturbed control plots that were not trenched. Data from four additional subplots, lined with 1 μm mesh as reported in [12], were not used here. In the present study we report two response variables. We define 'root-free fluxes' as those attributed to microbial activity using the subset of plots lined with 45 μm mesh. We also report fluxes attributed to roots alone calculated as Flux$_{undisturbed}$ - Flux$_{root\_free}$. For soil $CO_2$ flux this quantity is root respiration, and for $CH_4$ it is 'root-influenced' soil $CH_4$ as roots do not emit meaningful amounts of $CH_4$.

**Sap flow velocity.** Tree transpiration was measured to assess the impacts of the experimental floods on vegetation; tree sap flow velocity is a proxy for tree hydraulic function and physiological response to soil moisture and salinity (Table 1) [29]. Sap flow velocity was measured using the thermal dissipation method (Plant Sensors, Nakara, Australia; [30]). In this method two 3.5-cm-long probes with a 2-cm sensing length were installed horizontally and 10 cm apart in tree stems at ~1.7 m height. Both probes measure temperature but the top probe is also heated at a constant power while the bottom serves as a temperature reference. Sensors were installed in 2019 in 18 trees per plot constituting six replicates in each of three tree species, *Acer rubrum* (red maple), *Liriodendron tulipifera* (tulip polar), and *Fagus grandifolia* (American beech) and measured every 15-minutes continuously. Sap flow velocity (Fd), in cm/hr, was calculated based on [31] (Equations 1 & 2):

$$F_d = 118.99 \times 10^{-6} \left( \frac{dT_{max}}{dT} - 1 \right)^{1.231}$$

$$F_d = 360,000 \times F_d$$

where dT is the difference in the heated and reference probes and $dT_{max}$ is the maximum sap flow between the hours of 12:00 am and 5:00 am when sap flow is zero [30]. We only used daytime sap flow data (defined as the hours of 5:00 am

and 9:00 pm) for this analysis, to reduce the influence of inter-species differences in stem capacitance on transpiration [32]. Sap flow velocity was averaged by plot.

### Ecosystem-level analysis

Each variable was processed to include data between 20 June 2022 and 25 Jun 2022, except for porewater that used 13 Jun 2022 as a start date due to sampling frequency differences. These windows capture 48 hours before and after the experimental events. We only included data from TEROS sensors at 15 cm depth in the calculations for Equation 3 below to draw from the largest number of sensors and spatial variation across the plots. Analytes were kept at the collection frequency for each variable. To assess changes and test our hypotheses across heterogeneous data types, data were first binned into five distinct groups: 1) "Pre" (pre-events, prior to 22 Jun 2022 05:30 EST), 2) "Mid" (flood event, 22 Jun 2022 05:30–22 Jun 2022 14:30 EST), 3) "Between Events" (22 Jun 2022 14:30 EST – 22 Jun 2022 18:00 EST), 4) "Rain Event" (22 Jun 2022 18:00–23 Jun 2022 22:30 EST), and 5) "Post" (post-events, after 23 Jun 2022 22:30 EST). The rain event was a natural event that occurred < 24 hours after the end of the planned experiment. We then calculated the change in each response variable during the event as (Equation 3):

$$change = V_{dist} - V_{pre}$$

where $V_{pre}$ is the average variable value of the Pre time period (baseline) and $V_{dist}$ is the largest absolute value during the disturbance event, either the minimum (variable decreased during event) or maximum (variable increased during event) variable value during the Mid time period.

We calculated the minimum and maximum changes from Equation 3 that were observed across all variables in the control plot. We define responses greater or less than the control plot variability as outside of natural variance and therefore a response to flooding.

### Statistics

This is an unreplicated study at the plot level, which limits our ability to do certain statistical tests, such as a direct comparison of changes through time with simple parametric tests. As we are comparing a small window of time for this analysis (48 hours pre/post event), and due to different sampling frequencies across variables, we also could not use the before-after-control-impact (BACI) statistical framework for which the larger experiment was designed [23].

To test our hypotheses, we adapted statistical approaches from other system-level analysis research to assess the differences among response categories (i.e., system component groups outlined in Table 1) using analysis of variance (ANOVA) followed by Tukey-HSD post-hoc test when significant [33]. These statistical frameworks work well on unreplicated sampling designs with a large number of response variables, such as those following an unplanned disturbance event such as a hurricane [33]. To explore the differences in the change (immediate response during event) across the treatment plots (freshwater/saltwater) and system components (hydrological, biogeochemical, and vegetation), we conducted a two-way ANOVA (change ~ system component x plot). To assess differences in changes observed (immediate response during event) within each system component (e.g., for all variables within the hydrological component, etc) across all plots (control, freshwater, saltwater), we conducted a one-way ANOVA for each system component (change ~ plot).

All statistical tests were conducted in R version 4.4.0 [34]. Datasets were tested for normality (Shapiro-Wilk's test) and equal variance (Bartlett's test), and we rank-normalize data to achieve normal distributions prior to statistical testing when assumptions of normality and equal variance were not met. Wherever percent difference is reported, it was calculated based on plot averages across timepoints, and no statistics were performed on these percent difference values to avoid possible pseudo replication issues [35]. All data and analytical code to reproduce our results are available at [28].

## Results

### Treatments

Widespread saturation of the forest soils during the event was achieved through application of 263 m³ of freshwater and 267 m³ of brackish water (S1 Table), consistent with our system test using only freshwater on both plots the prior year [23]. During the experiment, the incoming water had a conductivity of 127±4 µS/cm for the freshwater plot, and 13,666±1,206 µS/cm for the saltwater plot (equivalent to 7.9±0.7 PSU). It began to rain several hours after the experimental flooding event ended, adding 2.8 cm of rainfall over 29 hours.

### Change among ecosystem components

We first examined overall responses to flooding for hydrological, biogeochemical, and vegetation variables as a change pre-event to mid-event, relative to changes during that same time period in the control plot (Fig 2). There were no differences in variables mid-event relative to their values pre-event (i.e., change) between the treatment plots (freshwater and saltwater, p=0.557). However, we did find that changes in variables mid-event relative to their values pre-event differed across system components (p<0.05). Statistical differences across the system components (hydrological, biogeochemical, and vegetation; p<0.01) showed a very weak interaction with plot (p=0.094). Post-hoc tests revealed that hydrologic variable responses were different from both biogeochemical and vegetation responses (p<0.05), with no significant difference between the changes in vegetation and biogeochemical variables.

We found significant differences in the change from pre- to mid-event (Equation 3) between plots for the hydrological variables (p<0.01), which post-hoc Tukey-HSD analysis indicated were driven by differences between the saltwater and the control plot (p<0.01) and the freshwater and the control plot (p=0.014). No statistical differences in the change from pre- to mid-event (Equation 3) were observed across plots for the biogeochemical (p=0.833) and vegetation (p=0.114) system components.

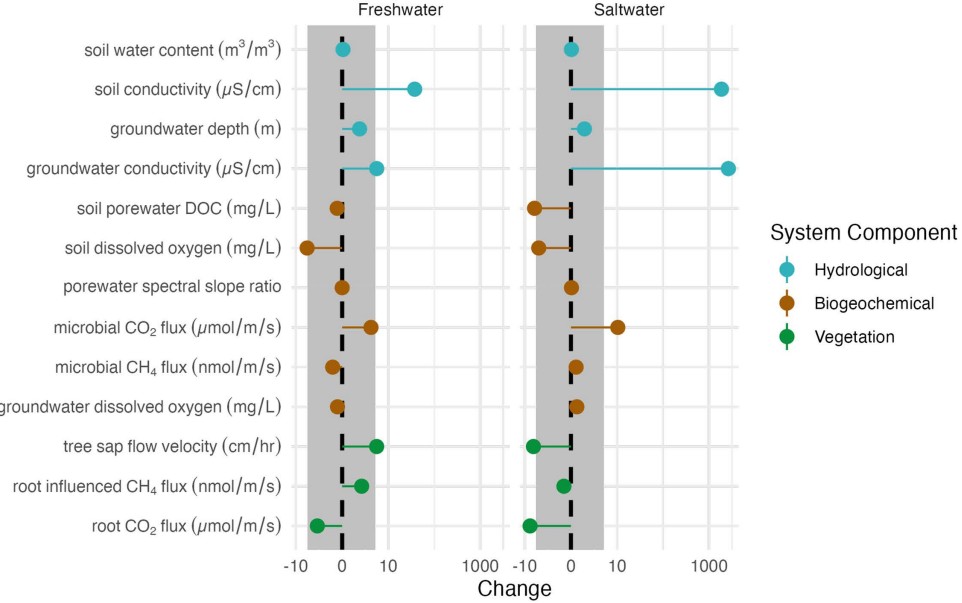

**Fig 2. Change in response variable mid-event compared to 48 hours prior (pre-event) for measured biogeochemical, hydrological, and vegetation variables (note log scale on x axis).** The grey shaded area indicates the change in control plot during the event for all variables during the study period.

## Hydrologic responses

Hydrologic variables had the largest immediate responses to the TEMPEST event in both treatment plots. Soil water content and groundwater depth did not change outside of the control plot variance window during the event (Fig 2). However, soil water content peaked toward the end of the treatment event for both the freshwater and saltwater plots and started to decrease immediately after the end of the treatment (Fig 3). During the event, soil water content in the freshwater plot increased ~20% from baseline, and ~10% from baseline in the saltwater plot (Fig 3). The groundwater level rose from a depth of ~3 m below the soil surface to within the root zone in both the saltwater (57 cm minimum depth) and freshwater plots (29 cm), peaking in both plots shortly after water application ended (S2 Fig).

An order of magnitude difference between the freshwater and saltwater plots for soil electrical conductivity at 15 cm persisted after the event throughout the soil profile (Fig 4). Soil electrical conductivity had the greatest overall response of all variables, with an increase that exceeded the control plot variance envelope in both the freshwater and saltwater plots (Fig 2). During the flooding event, conductivity at 15 cm depth increased by ~ 40% compared to baseline in the freshwater plot, and ~2700% compared to baseline in the saltwater plot (Fig 2 and S2 Fig). Groundwater electrical conductivity increased outside of the control plot variance envelope for both the freshwater and saltwater plots (Fig 2).

## Biogeochemical responses

Most biogeochemical parameters monitored in the soil, porewater, and groundwater responded similarly between freshwater and saltwater treatments relative to the control plot. Biogeochemical responses of small magnitude with varying directions of change from pre-event to mid-event include soil porewater spectral slope ratio, microbial $CH_4$ flux, and groundwater dissolved $O_2$ (Table 2). The spectral slope ratio (Sr), a metric of porewater DOC quality, responded weakly (i.e., within control plot variability) but in opposite directions, decreasing in the freshwater plot and increasing in the saltwater plot from pre-event to mid-event (Fig 2). Microbial $CH_4$ flux responded weakly relative to variance in the control plot and in different directions, with a decrease in the freshwater plot and an increase in the saltwater plot (Fig 2). The microbial fluxes from the saltwater plot root exclusion subplots momentarily shifted from being a methane sink (i.e., negative flux) to a source (i.e., positive flux to the atmosphere) during the flood treatment, whereas the root exclusion subplots in the freshwater and control plots remained $CH_4$ sinks throughout the experiment (S3 Fig). Finally, groundwater dissolved $O_2$ responses were lagged, noticeably increasing after the flooding event, and remaining elevated in the freshwater plot during the 48 hours following the event (S3 Fig).

Only three biogeochemical variables had changes pre-event to mid-event outside the control plot variance: soil dissolved oxygen, microbial $CO_2$ fluxes, and porewater DOC (Table 2). Soil $O_2$ at 30 cm was lower than control variance in the freshwater plot (Fig 2). Soil $O_2$ decreased in both the freshwater and saltwater plots during the event, reaching anoxia toward the end of the event in both treatment plots (Figs 2 and 3). Microbial $CO_2$ fluxes were higher than control variance in the saltwater plot. Interestingly, microbial $CO_2$ fluxes in the root exclusion subplots increased noticeably in the saltwater plot early on during the TEMPEST flooding event, resulting in a positive change during the event (Fig 2), and decreased toward the end of the event in both plots (S3 Fig). Porewater DOC had the most marked difference between the plots; – it was lower than control variance in the saltwater plot and did not exceed the control plot variance in the freshwater plot (Fig 2). However, the trends in soil porewater DOC concentrations were similar in both treatment plots and the control plot immediately following the flooding event (S3 Fig).

### Vegetation responses

Vegetation responses to our short-term treatments were not uniform across the treatment plots. Root respiration in the saltwater plot and tree sap flow velocity in both treatment plots were the only measured vegetation variables that exhibited changes pre-event to mid-event outside the control plot variance. Tree sap flow velocity followed consistent diel patterns throughout the flooding event (S4 Fig). Sap flow velocity exceeded control plot variance for freshwater and estuarine water plots; however, there were no clear patterns across plots. Root respiration was lowest during the flooding event in

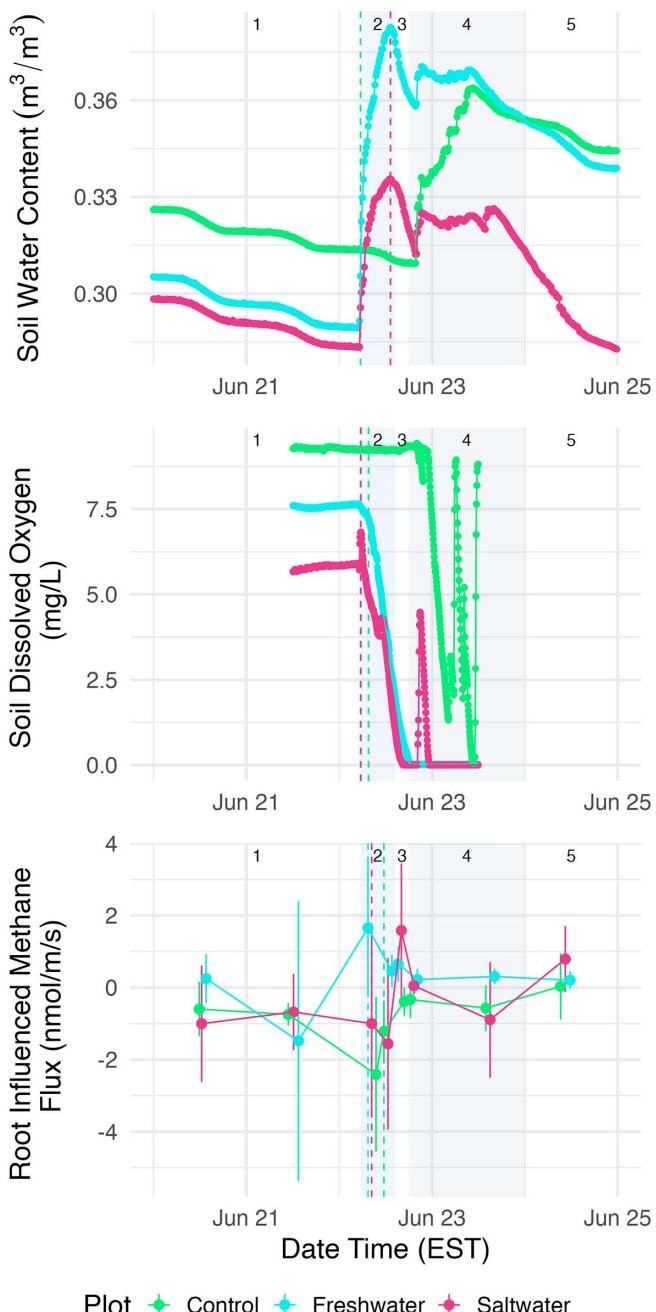

**Fig 3. Time series of example variable responses through study period for hydrological (soil water content), biogeochemical (soil O²), and vegetation (root-influenced methane) system components, 48 hours before, during, and 48 hours after the event.** Box 1 highlights the 48 hours before the flood event ("Pre"), box 2 is during ("Mid") the flooding event (blue), box 3 is after the flood event ("Between Events"), box 4 is the rain event ("Rain") following the flood event (grey), and box 5 is after the rain event is over ("Post"). Dashed lines in box 2 represent the Vdist used for the change calculations for each plot in the respective plot colors. Note that for soil O² the freshwater and saltwater Vdist are five minutes apart, and for soil water content freshwater and saltwater Vdist are at the exact same time stamp.

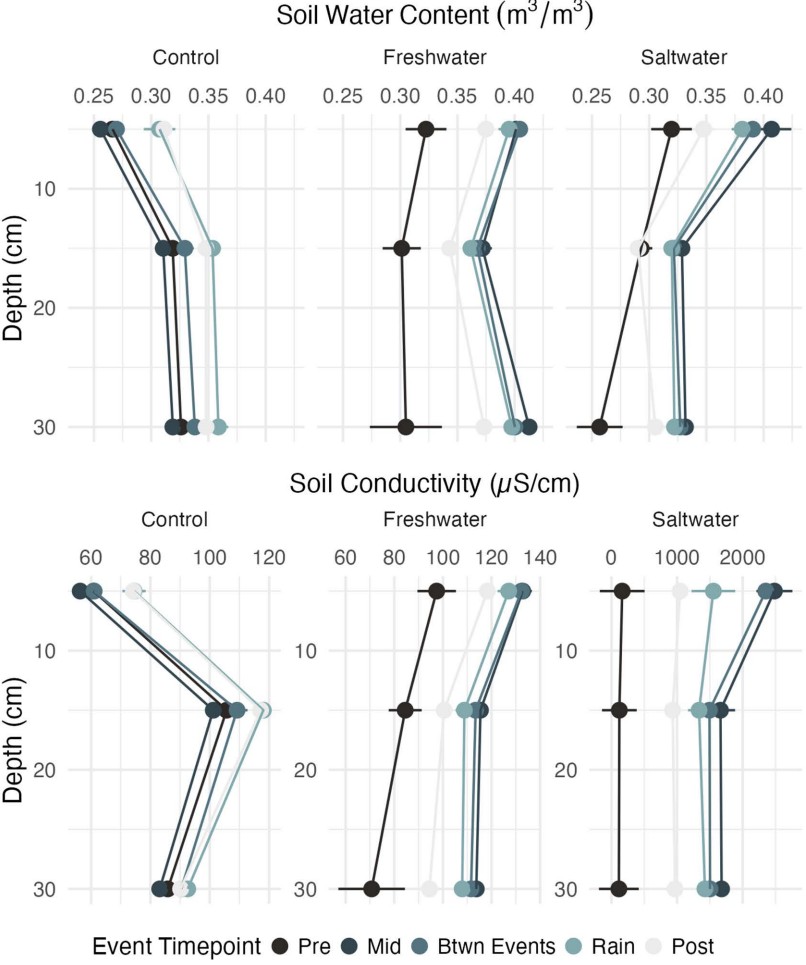

**Fig 4. Depth profiles of a) soil water content and b) soil conductivity for the 48 hours prior to the event (darkest shade), middle of flooding event, between flooding and rain event, during the rain event, and 48 hours after the rain event (lightest shade) for control (left), freshwater (middle) and saltwater (right) plots.** Note variable x axis on panel **b.**

both plots (Fig 2), but strong conclusions cannot be drawn because of its high variation (S4 Fig). Only the saltwater plot exceeded the envelope of control plot variance (Fig 2).

Root-influenced $CH_4$ fluxes measured in the root exclusion subplots did not exceed the envelope of control plot variance (Fig 2). However, root-influenced $CH_4$ fluxes peaked during the event in the freshwater plot and after the flooding event ("Between Events") in the saltwater plot (Fig 3). In the freshwater plot, root-influenced $CH_4$ fluxes shifted from an atmospheric sink (i.e., negative flux) to an atmospheric source during the flood treatment whereas the saltwater plot shifted to a source only momentarily to a source between events and post rain event (Fig 3).

## Natural rain event

The natural rain event had a noticeable impact on all hydrologic variables (Fig 3 and S2 Fig), soil $O_2$ (biogeochemical; Fig 3), and sap flow velocity (vegetation; S4 Fig). During the rain event, volumetric water content at 15 cm depth increased in all plots (Fig 3). Soil $O_2$ in both treatment plots remained lower than the pre-event values throughout the rain event and

**Table 2. Major system responses to the first TEMPEST event for each system component (48 hours pre, during, and post event).**

| Variable | System Component | Response to TEMPEST event |
|---|---|---|
| Soil water content | Hydrological | Short lived increase in both treatment plots **during** the event and rapid decrease after the event (Fig 3). |
| Soil conductivity | Hydrological | Changed outside control plot variance for freshwater and saltwater plots (Fig 2). Short-lived increase in the saltwater plot **during** the event and rapid decrease after the event (S2 Fig). |
| Groundwater depth | Hydrological | **Lagged** pulse increase in both treatment plots (S2 Fig). |
| Groundwater conductivity | Hydrological | Changed outside control plot variance for freshwater and saltwater plots (Fig 2). **Lagged** pulse increase in saltwater plots (S2 Fig). |
| Soil porewater dissolved organic carbon (DOC) | Biogeochemical | Changed outside control plot variance for saltwater plot (Fig 2). Decrease **during** the event in saltwater plot (S3 Fig). |
| Microbial methane ($CH_4$) flux | Biogeochemical | **Lagged** increase in saltwater plot. Sharp decline in freshwater plot **during** event, then subsequent increase (S3 Fig). |
| Microbial carbon dioxide ($CO_2$) flux | Biogeochemical | Changed outside control plot variance for saltwater plot (Fig 2). Sharp initial increase in saltwater plot **during** the event, followed by decline (S3 Fig). |
| Soil oxygen ($O_2$) | Biogeochemical | Changed outside control plot variance for freshwater plot. Steady decrease in both treatment plots **during** the event. Small pulse increases in saltwater plot at beginning of event (Fig 3). |
| Soil porewater spectral slope ratio | Biogeochemical | No large changes during the event in either plot but decrease in saltwater plot **following** the event (S3 Fig). |
| Groundwater dissolved oxygen ($O_2$) | Biogeochemical | **Lagged** pulse increase in both treatment plots (S3 Fig). |
| Tree sap flow velocity | Vegetation | Changed outside control plot variance for freshwater and saltwater plots (Fig 2). No clear patterns across plots (S4 Fig). |
| Root-influenced $CH_4$ flux | Vegetation | Increase **during** the event in freshwater plot, decrease then pulse increase in saltwater plot following the event (Fig 3). |
| Root $CO_2$ flux | Vegetation | Changed outside control plot variance for saltwater plot (Fig 2). Increase **during** event in freshwater plot, decrease in saltwater plot (S4 Fig). |

also decreased in the control plot during the rain event (Fig 3). Groundwater dissolved $O_2$ increased during the rain event in all plots (S3 Fig). Tree sap flow velocity was lower in all plots during the rain event (S4 Fig).

## Discussion

The punctuated hydrologic disturbance we applied to the ecosystem was the same order of magnitude as a tropical storm or hurricane in the region [23], and the increase in groundwater level observed in response to this event is analogous to increases in water table observed following hurricanes along the East Coast of the US [36]. Likewise, the increases we observed in groundwater salinity in the saltwater plot has been previously observed in coastal upland forests following a hurricane event [37]. We therefore consider our attempt to simulate the freshwater and saltwater hydrologic responses to a hurricane-scale event successful.

The single novel flooding event simulated in the first application of TEMPEST significantly perturbed several hydrologic parameters but, as expected, the effects were transient. We hypothesized the hydrologic effects of our first freshwater and saltwater treatments would be similar despite the potential for salts to alter soil properties [38]. This was supported by the rapid recovery of soil electrical conductivity in the 48 hours following the event across the saltwater plot. Although similar transient effects have been reported in other ecosystem-scale saltwater pulse-addition experiments [38], the rain event that occurred less than 24 hours after treatments ended likely accelerated the decline in soil conductivity by flushing out the added ions from the treatment waters (Fig 4). In the absence of a rain event we expect that electrical conductivity (a proxy for salinity) would have remained elevated for a longer period of time [39,40]. Other studies have found no significant effect of freshwater additions on soil electrical conductivity [41,42]. We attribute the observed increase in soil

electrical conductivity in the freshwater plot to enhanced ion mobility (e.g., increased electromigration) associated with higher soil water content rather than an increase in salinity. It is worth noting that the municipal freshwater source used for the experiment has a slightly elevated electrical conductivity (127 µS/cm) in relation to pre-event soil conductivity, which contributed to the observed increase in the measured electrical conductivity in the freshwater plot. However, the elevated electrical conductivity in the freshwater treatment was far too low (0.06 PSU) to cause stress effects (< 0.5 PSU is generally considered to be freshwater [43]).

Although the hydrologic impacts of the first event were short-lived, even transient changes in soil saturation and salinity can perturb soil biogeochemical processes and plant physiology through shifts in soil $O_2$ and changes in soil ionic strength [36,44]. The relative decline in soil $O_2$ was surprising for a short-term event and suggests that this upland forest, with typically aerobic soils, may shift quite rapidly toward anaerobic biogeochemical processes as the duration of events lengthens from several hours to several days [14]. The larger decline in porewater DOC in the saltwater plot compared to the freshwater plot during the event may have been driven by estuarine salts reducing carbon solubility in porewater solutions [22], an effect that is expected to impact microbial activity, soil organic matter decomposition, greenhouse gas fluxes, and hydrologic export of DOC to the adjacent estuary if salts accumulate in the plot's soils [22,45,46]. DOC chemical composition based on the spectral slope ratio responded very weakly and in different directions in the two treatments. The slight increase in spectral slope ratio in the saltwater plot suggests preferential removal of higher molecular weight compounds from the DOC pool during saline flooding, while freshwater flooding may retain these compounds [47]. This change in DOC quality may influence the availability of the porewater DOC to be utilized by microbial communities, as a higher slope ratio is usually associated with more degraded, lower-molecular weight organic matter [47]. Despite the fact that over half of the variables examined exceeded the native variability of the control plot (7 of 13; Fig 2), the treatment responses were generally either inconsistent in direction or small in overall magnitude. However, subtle differences in most biogeochemical variables, even those not outside of the control plot variance, may still be relevant for biogeochemical functioning in the system.

The short-term response of soil and groundwater salinity and $O_2$ observed here likely explain the lack of large or sustained shifts in above and belowground biogeochemical processes to the novel treatments. This is consistent with the results of freshwater tidal marsh experiments where novel exposure to saline water added as a pulse caused relatively weak biogeochemical and vegetation responses compared to press disturbances [15,38,41]. Soil flooding events that occur during and after natural hurricanes often last much longer than our event, on the order of days, leading to longer-duration changes in soil $O_2$ and salinity, and stronger impacts on biogeochemical cycles [48]. The muted responses may also reflect high within-variable variation, relatively high spatial variability in the control plot, or both (Figs 3 and S2–S4 Fig). We acknowledge that heterogeneity within a plot of hydrology and soil properties influence our ability to understand causal relationships between driver and response variables of tree mortality.

The long-term implications of the transient changes in $CO_2$ and $CH_4$ fluxes observed in the first 48-hours following flooding are harder to discern because they are determined by interacting physical and biological factors and display high spatial variability throughout the study period. The brief increase we observed in root-free (heterotrophic) soil $CO_2$ emissions may be due to physical displacement of $CO_2$ from the soil matrix [49,50]. $CH_4$ uptake by soils is strongly influenced by soil moisture content, as higher water content limits $CH_4$ diffusion from the atmosphere into the soil, and constrains microbial utilization [51–53]; the higher water content in the freshwater plot could therefore explain the lower $CH_4$ response compared to the saltwater plot. Such physical factors are likely to dominate greenhouse gas responses over short time scales (hours), while biogeochemical and vegetation contributions become increasingly important to net greenhouse gas responses as the frequency or duration of events increase [54].

The increases in soil saturation and decreases in oxygen availability that may have influenced soil greenhouse gas fluxes can also influence vegetation responses, including root functions and tree physiology. Although vegetation structural responses would certainly be outside of our 48-hour response window, tree sap flow velocity and root greenhouse gas fluxes are governed by immediate physical and physiological constraints on water and oxygen availability that may

respond within that timeframe [9,12]. Therefore, we examined the responses of tree sap flow velocity, root respiration, and root-influenced soil $CH_4$ flux, with the understanding that the influence of roots on $CH_4$ flux would operate indirectly through physical or microbial processes. Sap flow responded in opposite directions in the two treatments. Freshwater addition increased sap flow suggesting that transpiration was water-limited, a possibility that aligns with evidence of water limitation of forest net primary production at the site [55]. The decrease in sap flow when the added water was saline suggests a stress response [9]. The decrease in root respiration in the freshwater and saltwater plots suggests a stress response that may have reduced root respiration rates [56,57]. Together, these results confirm that short-term hydrologic events can induce measurable changes in plant physiological functions, with freshwater additions temporarily removing water limitation and salinity exposure potentially inducing stress responses. However, the large variation in root respiration within each plot (S4 Fig) argues for additional observations to establish belowground cause and effect mechanisms.

Transient changes in key drivers of tree mortality, such as soil moisture and oxygen levels, can influence long-term responses by changing baseline conditions, affecting the ability of the system to respond to future perturbations [24]. Evidence for such legacy effects is suggested by the response of the system to a rain event that occurred < 24 hours after the TEMPEST treatments ended. Soil $O_2$ concentrations in the saltwater plot had only partially recovered (to about half of atmospheric saturation) before rain began, while concentrations in the control plot started at full saturation. Consequently, soil $O_2$ concentrations were depleted much faster in the saltwater than control plots post-event, consistent with the differences in initial conditions due to the flooding event. While this single event cannot confirm causality, it suggests that even the brief anaerobic conditions due to the flooding event could alter antecedent conditions in ways that influence responses to subsequent disturbances. Repeated episodic events may amplify these effects, potentially driving changes in belowground redox, carbon, and nutrient cycling that may become more pronounced with repeated episodic events [58,59] and inducing shifts in key processes regulating vegetative dynamics leading to plant stress and mortality [9].

## Conclusions and future research directions

This study detailed the first responses of an upland coastal forest to a field-scale flooding experiment designed to decouple two distinct disturbances associated with extreme, hurricane-scale storm events: (1) flooding from heavy precipitation and (2) exposure to saline conditions from storm surge [23]. The goal of this study was to identify early indicators of change in hydrologic and biogeochemical mechanisms that impacts plant ecophysiology driving ghost forest creation [9]. We observed transient responses in key drivers of plant stress – soil salinity and dissolved oxygen – even though there was no clear indication of plant stress from our vegetation variables. Likewise, we found changes in soil porewater DOC concentrations – a key driver of soil microbial functions – even though there was no clear indication of changes in microbially driven greenhouse gas fluxes.

While the ecosystem appears to have been resilient to one novel flooding event in the short-term (48 hours after), the rapid development of these responses illustrates how such events can stimulate immediate responses spanning the coupled above and belowground ecosystem, establishing novel antecedent conditions that influence ecosystem sensitivity to subsequent perturbations [24]. Future research coupling large-scale field manipulation with model simulations [60] and laboratory experiments [11] will be key to further disentangling mechanistic relationships and generate predictive understanding [61]. As flooding frequency and duration increase in coastal landscapes, this study's results help to identify which short-term responses arise and are most likely to accumulate into long-term alterations in ecosystem structure and functions. Such information will be crucial to understand and predict coastal forest vulnerability to changing environmental conditions expected throughout the 21st century [61].

## Supporting information

**S1 Table. Water application for the TEMPEST experiment.**
(DOCX)

**S1 Fig. Plot installation locations.**
(PNG)

**S2 Fig. Hydrological variable responses (soil conductivity, groundwater depth, groundwater conductivity) through time.**
(PNG)

**S3 Fig. Biogeochemical variable responses (porewater DOC, microbial $CH_4$ flux, microbial $CO_2$ flux, porewater slope ratio, and groundwater DO) through time.**
(PNG)

**S4 Fig. Vegetation variable responses (sap flow velocity, root carbon dioxide flux) through time.**
(PNG)

## Acknowledgments

We thank the massive field sampling and infrastructure teams that made this first TEMPEST event possible – in particular, Rick Smith, Stella Woodard and Evan Phillips. The full list of TEMPEST 1.0 event samplers are listed in the consortium authorship. We also thank James Stegen for his contributions to early conceptualization of the TEMPEST experimental design. The Pacific Northwest National Laboratory is operated for DOE by Battelle Memorial Institute under contract DE-AC05–76RL01830.

**TEMPEST 1.0 event consortium authors**

Pat Megonigal[1]*, Anya Hopple[1], Rick Smith[2], Alice Steans[1], Evan Phillips[1], Stella Woodard[2], Stephanie Pennington[3], Ben Bond-Lamberty[3], Nate McDowell[4], Gary Peresta[1], Roy Rich[1], Kendal Morris[3], Donnie Day[5], Wei Huang[6], Mia DiCianna[1], Lani DuFresne[1], Sam Wright[1], Allison Myers-Pigg[7], Opal Otenburg[7], Khadijah Homolka[7], Madison Bowe[7], Nicholas Ward[7], Roberta Peixoto[5], Jonathan Kwong[1], Peter Regier[7], Elaine Yu[1], Kennedy Doro[5], Moses Adebayo[5], Emmanuel Efemena[5], Solomon Ehosioke,[5] Mitchell Smith[3], Yerang Yang[1], Drew Peresta[1], J. Alan Roebuck Jr.[7], Leticia Sandoval[5]

1 Smithsonian Environmental Research Center, Edgewater, MD

2 Global Aquatic Research, LLC, Sodus, NY

3 Joint Global Change Research Institute, Pacific Northwest National Laboratory, College Park, MD

4 Pacific Northwest National Laboratory, Richland, WA

5 University of Toledo, Toledo, OH

6 Oak Ridge National Laboratory, Oak Ridge, TN

7 Pacific Northwest National Laboratory, Sequim, WA

* consortium lead author: megonigalp@si.edu

## Author contributions

**Conceptualization:** Allison Myers-Pigg, Anya Hopple, Stephanie C. Pennington, Peter Regier, Ben Bond-Lamberty, Kennedy O. Doro, Nate McDowell, Nicholas D. Ward, Vanessa L. Bailey, J. Patrick Megonigal.

**Data curation:** Allison Myers-Pigg, Anya Hopple, Stephanie C. Pennington, Peter Regier, Mia J. DiCianna, Kennedy O. Doro, Julia McElhinny.

**Formal analysis:** Allison Myers-Pigg, Stephanie C. Pennington, Peter Regier, Mia J. DiCianna, Julia McElhinny.

**Funding acquisition:** Ben Bond-Lamberty, Nicholas D. Ward, Vanessa L. Bailey, J. Patrick Megonigal.

**Project administration:** Allison Myers-Pigg, Anya Hopple, Stephanie C. Pennington, Ben Bond-Lamberty, Kennedy O. Doro, Nate McDowell, Alice Stearns, Nicholas D. Ward, Vanessa L. Bailey, J. Patrick Megonigal.

**Resources:** Allison Myers-Pigg, Ben Bond-Lamberty, Kennedy O. Doro, Nicholas D. Ward, Vanessa L. Bailey, J. Patrick Megonigal.

**Software:** Allison Myers-Pigg, Anya Hopple, Stephanie C. Pennington, Peter Regier, Julia McElhinny.

**Supervision:** Allison Myers-Pigg, Anya Hopple, Stephanie C. Pennington, Ben Bond-Lamberty, Kennedy O. Doro, Nate McDowell, Nicholas D. Ward, J. Patrick Megonigal.

**Validation:** Allison Myers-Pigg, Anya Hopple, Stephanie C. Pennington, Peter Regier.

**Visualization:** Allison Myers-Pigg, Stephanie C. Pennington, Peter Regier, Alice Stearns.

**Writing – original draft:** Allison Myers-Pigg, Stephanie C. Pennington, Peter Regier, J. Patrick Megonigal.

**Writing – review & editing:** Allison Myers-Pigg, Anya Hopple, Stephanie C. Pennington, Peter Regier, Ben Bond-Lamberty, Mia J. DiCianna, Kennedy O. Doro, Nate McDowell, Julia McElhinny, Alice Stearns, Nicholas D. Ward, Vanessa L. Bailey, J. Patrick Megonigal.

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
