## [Decision Letter · Decision Letter 0]

8 Oct 2025

PONE-D-25-19393Short-term coastal forest responses to a hurricane-scale freshwater and saltwater flooding experimentPLOS ONE

Dear Dr. Myers-Pigg,

Thank you for submitting your manuscript to PLOS ONE. After careful consideration, we feel that it has merit but does not fully meet PLOS ONE’s publication criteria as it currently stands. Therefore, we invite you to submit a revised version of the manuscript that addresses the points raised during the review process.

**ACADEMIC EDITOR:** **Dear authors, thank you considering PLOS One. The study is interesting and within the scope of the Journal. However, based on the reviewers comments, the manuscript needs substancial changes. Please refer to the comments of each reviewer and address all queries.**

We look forward to receiving your revised manuscript.

Kind regards,

Miguel Inácio

Academic Editor

PLOS ONE

“COMPASS-FME, a multi-institutional project supported by the U.S. Department of Energy, Office of Science, Biological and Environmental Research as part of the Environmental System Science Program.”

“We thank the massive field sampling and infrastructure teams that made this first TEMPEST event possible – in particular, Rick Smith, Stella Woodard and Evan Phillips. We also thank James Stegen for his contributions to early conceptualization of the TEMPEST experimental design. This research was supported by COMPASS-FME, a multi-institutional project supported by the U.S. Department of Energy, Office of Science, Biological and Environmental Research as part of the Environmental System Science Program, and by the Smithsonian Environmental Research Center. The Pacific Northwest National Laboratory is operated for DOE by Battelle Memorial Institute under contract DE-AC05-76RL01830.”

“COMPASS-FME, a multi-institutional project supported by the U.S. Department of Energy, Office of Science, Biological and Environmental Research as part of the Environmental System Science Program.”

6. One of the noted authors is a group or consortium [TEMPEST 1.0 Event Consortium]. In addition to naming the author group, please list the individual authors and affiliations within this group in the acknowledgments section of your manuscript. Please also indicate clearly a lead author for this group along with a contact email address.

7. We note that Figure 1 in your submission contain copyrighted images. All PLOS content is published under the Creative Commons Attribution License (CC BY 4.0), which means that the manuscript, images, and Supporting Information files will be freely available online, and any third party is permitted to access, download, copy, distribute, and use these materials in any way, even commercially, with proper attribution. For more information, see our copyright guidelines: http://journals.plos.org/plosone/s/licenses-and-copyright.

Reviewers' comments:

Reviewer's Responses to Questions

**Comments to the Author**

1. Is the manuscript technically sound, and do the data support the conclusions?

Reviewer #1: Yes

Reviewer #2: No

Reviewer #3: Partly

2. Has the statistical analysis been performed appropriately and rigorously? 

Reviewer #1: I Don't Know

Reviewer #2: No

Reviewer #3: Yes

3. Have the authors made all data underlying the findings in their manuscript fully available?

Reviewer #1: Yes

Reviewer #2: Yes

Reviewer #3: Yes

4. Is the manuscript presented in an intelligible fashion and written in standard English?

Reviewer #1: Yes

Reviewer #2: Yes

Reviewer #3: No

5. Review Comments to the Author

Reviewer #1: Summary

This manuscript reports data collected during early stages of the TEMPEST project which aims to better understand mechanisms driving upland forest conversion through fresh and saltwater flooding events that are increasing in frequency. They measure a suite of hydrologic, biogeochemical and vegetation variables that can indicate change in ecosystem function before trees are visibly dying and therefore may provide a greater understanding of the mechanisms that result in ghost forest formation and marsh encroachment. Although authors provide motivation for the larger study, it is not clear what can be learned from looking at the data presented here which was collected after the first of many planned flooding events. More specifics could be included in the introduction to motivate this study in isolation from the larger TEMPEST project to make it a more compelling manuscript. Below I provide general comments followed by more specific line-by-line comments about this overarching concern.

General comments

Abstract

• It would be helpful to include some indication of the flooding treatment that you are referring to and how it allows you to decouple the impacts of freshwater flooding and saltwater exposure on upland forests.

• The final sentence of the abstract reflects a larger issue with the manuscript. Beyond just suggesting that fore frequent/sustained flooding events are needed to drive more significant change, what can be learned the lack of change in many measured variables in response to this first flooding event?

Introduction

• The introduction could use some more specific information to provide context for your hypotheses and the suite of measurements included in this study. Some examples of missing information I think could improve the introduction:

o L59: What are the ‘hydrologic extremes’ you are referring to and what is driving increases in the frequency of their occurrence?

o What are the broader implications of studying hydrologic shifts and upland forest conversion? Why is this study important (beyond the general statement about improving modelling)?

• Could be coming at this from the angle of carbon storage, greenhouse gas production, habitat availability, etc…

o What specific shifts in belowground biogeochemistry are expected and how might those lead to shifts in vegetation? There is currently no context for the specific measurements mentioned in Table 1 and outlined in the methods section.

o L72-80: Could be replaced with a more specific explanation of the gaps in knowledge that past studies have not yet filled. This paragraph is currently very vague.

o Clarify the importance of separating the effects of freshwater and saltwater flooding?

o Hypothesis statements are also very broad. Would it be possible to have the hypothesis statements more directly address specific mechanisms related to the potential impacts of flooding a system for the first time (as opposed to chronic flooding that will be tested later in the experiment)? Are there specific metrics that are expected to change early in the experiment and therefore act as early signals of forest conversion?

Discussion

• Similar concern as was expressed for the abstract and introduction: most of the discussion is focused on transient or non-significant changes that could be more important with increased duration/frequency of flooding later in the experiment. Did we learn anything about forest conversion from looking at results from the first flooding event in isolation?

Line by line comments:

• L60: might be helpful to indicate that formation of ghost forests is driven by tree mortality

• L98: This is the first time you mention pulse and press disturbance. I would suggest removing this terminology or describing the concept in more detail.

• L81-82: I would suggest providing some indication of your experimental design here given that you have highlighted that your methods are novel and therefore an important part of understanding the importance of your work.

• L74-76: What are you referring to when you say “this is particularly true”? I would suggest being more specific. What types of shifts in belowground biogeochemistry do you expect to observe?

• L82-85: Specify the timescale are you referring to in this hypothesis.

• L 89: What do you specifically mean by ‘experimental flooding’? This might be a good place to very briefly introduce the flooding treatments used in your study.

• Figure 1: Provide more explanation of the blue line and boxes in Figure 1b. What is the frequency of water pulse events and does it change through time? Clarify that data in this ms is coming from after the first flooding event (blue box 1?), and future work will assess how effects of flooding change with subsequent flooding events. Is ‘Unseen ecological responses’ indicating a hypothesis or result?

• L131: How frequent are the flooding events?

• L191-197: When were these gas measurements collected in relation to the flooding event. How frequently?

• L 207-208: There is currently no context for understanding the potential for roots to influence CH4 fluxes.

• L250-251: How are you assessing differences in flooding response between fresh and saltwater plots when this is an unreplicated experiment?

• L290-292: This statement is confusing – consider revising to clarify the takeaway from this finding.

• Figure 4: Not intuitive to see x axes on the top rather than the bottom of the figures.

• Table 2: Seems like this could be better shown in a graph rather than table.

• Lines 363-364, 368-369, etc (throughout results): Should changes that are smaller than variability measured in the control plots be reported as change? It seems more accurate to say that neither saltwater nor freshwater flooding drove significant change in porewater DOC/microbial CH4 flux….

• L374 (and Table 2): The first time ‘lagged’ terminology is introduced. It would help to have context for understanding why timing of responses to flooding may differ between variables and why this could be important?

• L494-496: I see this concept underlying the importance of your current study (i.e. responses of an upland forest to a single flooding event) and could be introduced much earlier and more explicitly.

• L502-504: A shortcoming of the study that should be acknowledged as such.

• L509-524: Not sure this paragraph belongs in the discussion section because it doesn’t include any discussion of results being presented in this manuscript and instead is just plugging the importance of the larger project.

Reviewer #2: The authors present an experiment designed to test the effects of a pulse disturbance on hydrologic, biogeochemical, and vegetation components in a coastal upland forest. While the study addresses an important topic, it suffers from several significant weaknesses that prevent it from being suitable for publication in its current form.

One of my primary concerns is the lack of replication in the experimental design, which severely undermines the validity of the results and the reliability of the conclusions drawn. Without replication, it is difficult to distinguish treatment effects from background variability. Secondly, the scientific relevance of the experiment is questionable. As the authors rightly point out in the introduction, we are facing a context of rapidly changing climate. Given this, an experimental design that investigates high-frequency or chronic flooding events would be of greater relevance and value to the scientific community than a single pulse disturbance. I would strongly recommend rethinking the experimental setup to test different climate change scenarios, particularly in relation to the increasing frequency and intensity of storm surges.

For these reasons, I regret to recommend rejection of this manuscript for publication.

Reviewer #3: Manuscript Title: Short-term coastal forest responses to a hurricane-scale freshwater and saltwater flooding experiment

In this research, the authors studied freshwater and saltwater flooding in the coastal region. The study is important but needs extensive revision before publication.

1. Title: Please check if the “TEMPEST 1.0 Event Consortium” is suitable for the author list?

2. Abstract: The method and main results of the entire study are missing. The author needs to add more methods and applied results to understand the actual study analysis.

3. “Most studies on the conversion of coastal upland ecosystems to wetlands are observation-based, using a space for time substitution design that is ultimately limited in its ability to test hypotheses about the mechanisms that underlie ecosystem state change.” Need reference.

4. “vegetation community structures” is this an appropriate word? Recheck this.

5. The introduction section lacks an in-depth discussion of the current study, research method and literature review.

6. Research novelty is also limited in understanding. Restructure the introduction section.

7. “We delivered 263 m3 of freshwater and 267 m3” try to improve your result sections.

8. In-depth discussion of the result section is very limited.

9. Limitations and future research directions need to be added after discussion.

10. The conclusion is missing in the study.

The manuscript needs more attention and revision accordingly. Based on the current status of the manuscript, I must recommend extensive major revision before publication.

6. PLOS authors have the option to publish the peer review history of their article (what does this mean?). If published, this will include your full peer review and any attached files.

Reviewer #1: No

Reviewer #2: No

Reviewer #3: No

---

## [Author Response · Author response to Decision Letter 1]

10 Nov 2025

Please see the detailed response to reviewer document in the file attachments.

---

## [Decision Letter · Decision Letter 1]

13 Jan 2026

PONE-D-25-19393R1Short-term coastal forest responses to a hurricane-scale freshwater and saltwater flooding experimentPLOS One

Dear Dr. Myers-Pigg,

Thank you for submitting your manuscript to PLOS ONE. After careful consideration, we feel that it has merit but does not fully meet PLOS ONE’s publication criteria as it currently stands. Therefore, we invite you to submit a revised version of the manuscript that addresses the points raised during the review process.

Dear authors, Thank you for your efforts in addressing the reviewers comments previously.

Please refer to the comments of both reviewers, specially Reviewer 1, and address all the comments and suggestions.==============================

We look forward to receiving your revised manuscript.

Kind regards,

Miguel Inácio

Academic Editor

PLOS One

Journal Requirements:

Reviewers' comments:

Reviewer's Responses to Questions

**Comments to the Author**

1. If the authors have adequately addressed your comments raised in a previous round of review and you feel that this manuscript is now acceptable for publication, you may indicate that here to bypass the “Comments to the Author” section, enter your conflict of interest statement in the “Confidential to Editor” section, and submit your "Accept" recommendation.

Reviewer #1: (No Response)

Reviewer #3: All comments have been addressed

2. Is the manuscript technically sound, and do the data support the conclusions?

Reviewer #1: Partly

Reviewer #3: Yes

3. Has the statistical analysis been performed appropriately and rigorously? 

Reviewer #1: Yes

Reviewer #3: Yes

4. Have the authors made all data underlying the findings in their manuscript fully available?

Reviewer #1: Yes

Reviewer #3: Yes

5. Is the manuscript presented in an intelligible fashion and written in standard English?

Reviewer #1: Yes

Reviewer #3: Yes

6. Review Comments to the Author

Reviewer #1: (No Response)

Reviewer #3: The authors have revised their work properly in response to the previous comments, but some minor modifications are still needed for this manuscript.

1. Why are references added in the conclusion section? The conclusion must be from your own study and analysis results.

2. If those references are for future recommendations, please separate the future recommendations and conclusion.

Best of Luck

7. PLOS authors have the option to publish the peer review history of their article (what does this mean?). If published, this will include your full peer review and any attached files.

Reviewer #1: No

Reviewer #3: No

---

## [Author Response · Author response to Decision Letter 2]

9 Feb 2026

Please see the response to reviewers letter for responses to all editorial comments.

---

## [Decision Letter · Decision Letter 2]

31 Mar 2026

PONE-D-25-19393R2Short-term coastal forest responses to a hurricane-scale freshwater and saltwater flooding experimentPLOS One

Dear Dr. Myers-Pigg,

Thank you for submitting your manuscript to PLOS ONE. After careful consideration, we feel that it has merit but does not fully meet PLOS ONE’s publication criteria as it currently stands. Therefore, we invite you to submit a revised version of the manuscript that addresses the points raised during the review process.

**ACADEMIC EDITOR:**  **Dear authors, thank you for the thorough review in response to the previous reviewers' comments.**

**Please refer to Reviewer 1's comments.**

We look forward to receiving your revised manuscript.

Kind regards,

Miguel Inácio

Academic Editor

PLOS One

Journal Requirements:

Reviewers' comments:

Reviewer's Responses to Questions

**Comments to the Author**

1. If the authors have adequately addressed your comments raised in a previous round of review and you feel that this manuscript is now acceptable for publication, you may indicate that here to bypass the “Comments to the Author” section, enter your conflict of interest statement in the “Confidential to Editor” section, and submit your "Accept" recommendation.

Reviewer #1: (No Response)

Reviewer #3: All comments have been addressed

2. Is the manuscript technically sound, and do the data support the conclusions?

Reviewer #1: Yes

Reviewer #3: Yes

3. Has the statistical analysis been performed appropriately and rigorously? 

Reviewer #1: Yes

Reviewer #3: Yes

4. Have the authors made all data underlying the findings in their manuscript fully available?

Reviewer #1: Yes

Reviewer #3: Yes

5. Is the manuscript presented in an intelligible fashion and written in standard English?

Reviewer #1: Yes

Reviewer #3: Yes

6. Review Comments to the Author

Reviewer #1: Summary

Authors edits in were extremely thorough and I found this version of the manuscript to be significantly improved. Some final thoughts summarized below:

For the introduction, I think there is some room to clarify specific sentences (called out in line by line comments), but generally, the study is effectively introduced!

For the discussion, I appreciated how the topic sentences of each paragraph made the flow of the story really clear! I called out some specific spots in the discussion where I think that 1) results could reiterated or 2) interpretation/synthesis could be stronger.

Line by line comments:

Lines 67-70: Long and some confusing structure – could be revised to increase clarity

L72: I think this statement could be strengthened by clarifying changes in what?

L 87: Clarify what you mean by “initial soil properties”. Are you talking about prior exposure to flooding/salinity?

L96-100: I like the idea of this sentence, but it reads awkwardly and ends up being difficult to follow. Maybe rephrase?

L108-113: I think this hypothesis statement could be condensed to make your meaning more clear.

L454-455: Might be worth a brief explanation of ‘electromigration’ because this is the only time it is mentioned in the paper. As a non-expert, Im not sure if this is an important takehome or not!

L468-471: Awkward structure – consider editing. Additionally, Im wondering if there is any interpretation of this finding (saltwater inundation decreases concentrations of high molecular weight compounds in DOC, but freshwater inundation does the opposite)?

L473: Not sure what you mean by “quantitatively important anaerobic biogeochemical cycles” – the meaning of this interpretation could be clarified.

L480-505: Final sentences in these two paragraphs negates any of your findings – kindof deflates any takeaway points. I wonder if the structure could be shifted so that you acknowledge shortcomings, and then highlight important takeaways despite these shortcomings?

498-499: I would suggest spelling out mechanism more explicitly – say the relationship between soil moisture content and CH4 diffusion rather than just mentioning that there is a relationship.

L500: I wouldn’t assume that readers remember differences in CH4 responses between plots. This is actually a general comment about the discussion. In some cases, think I could be helpful to add reminders about general findings before diving into interpretation.

L524-520: Is there any take-home message from these results? How should readers be thinking about the potential for pulses of fresh/saltwater to affect veg?

L523-532: Im not overly convinced that there was enough information gathered from the rain event to support your ideas here. Maybe ideas just need to be clarified further?

Reviewer #3: The authors are revised the manuscript based on the comments. Therefore, I must recommend accept manuscript for publication

7. PLOS authors have the option to publish the peer review history of their article (what does this mean?). If published, this will include your full peer review and any attached files.

Reviewer #1: No

Reviewer #3: No

---

## [Author Response · Author response to Decision Letter 3]

2 Apr 2026

Please see response to reviewers document where we address all reviewer comments in detail.

---

## [Editor Report · Decision Letter 3]

8 Apr 2026

Short-term coastal forest responses to a hurricane-scale freshwater and saltwater flooding experiment

PONE-D-25-19393R3

Dear Dr. Myers-Pigg,

We’re pleased to inform you that your manuscript has been judged scientifically suitable for publication and will be formally accepted for publication once it meets all outstanding technical requirements.

Kind regards,

Miguel Inácio

Academic Editor

PLOS One
---

## [Editor Report · Acceptance letter]

PONE-D-25-19393R3

PLOS One

Dear Dr. Myers-Pigg,

I'm pleased to inform you that your manuscript has been deemed suitable for publication in PLOS One. Congratulations! Your manuscript is now being handed over to our production team.

Kind regards,

on behalf of

Dr. Miguel Inácio

Academic Editor

PLOS One